# The Walking Trail Making Test is more accurate than a Dual-Task Walking Test for screening the level of fall risk among community-dwelling older people

Rafael Mauti[1], Romain Tisserand [2,3], Anaïck Perrochon[4], Melanie Gallot[1], Arnaud Peronin[5], Thomas Gilbert[6,7], Patrick Fargier[1,8☯], Pascal Chabaud [1☯*]

1 LIBM, EA 7424, Université Lyon 1, Université Jean Monnet, Univ. Savoie Mont Blanc, Villeurbanne, France, 2 PPRIME, Université de Poitiers, ISAE-ENSMA, CNRS, Poitiers, France, 3 CeRCA, Université de Poitiers, Université de Tours, CNRS, Poitiers, France, 4 HAVAE, UR 20217, Université de Limoges, Limoges, France, 5 Ligue Auvergne Rhône-Alpes de Rugby à XIII, Fédération Française de Rugby à XIII, Lyon, France, 6 RESHAPE, UMR S1290, Université Lyon 1, INSERM, Lyon, France, 7 Hospices Civils de Lyon, Hop Lyon Sud, Université Lyon 1, Pierre-Bénite, France, 8 University of Teacher Education - State of Vaud (HEPVD), Lausanne, Switzerland

☯ These authors contributed equally to this work.
* pascal.chabaud@univ-lyon1.fr

## Abstract

### Background

Preventing falls is a major issue for older people. Aging leads to a decline in executive functions and functional capacities, increasing the risk of falling. Most cognitive-motor assessments are *additional* Dual-Tasks (DTs), open to prioritization strategies, leading to heterogeneous results in the screening of the level of fall risk. The Walking Trail Making Test (W-TMT) is an *incorporated* cognitive-motor task sensitive to detecting cognitive impairments. A high level of fall risk is generally associated with higher cognitive impairments. The objective of this study is to evaluate the detection performance of the most complex condition of the W-TMT (WTMT-B) in the screening of people at a high level of fall risk, and to compare it to an *additional* DT, the 6-Meter Walking Test in the DT condition (6MWT-DT).

### Methodology

101 community-dwelling older people participated in the study. They were classified into three levels of fall risk (i.e., low, moderate, high) by the French Health Examination Center Fall Risk screening tool. They performed the three conditions of the W-TMT (N, A, and B) and the 6MWT in single-task and DT conditions. For both tasks, execution times were measured and compared between the levels of fall risk. Concordance index (*c*-index) analysis was performed to evaluate and compare the overall classification performance of the W-TMT-B and the 6MWT-DT for the three levels

**Data availability statement:** All relevant data are available at the following address: https://doi.org/10.5281/zenodo.19665098.

**Funding:** The authors received no specific funding for this work.

**Competing interests:** The authors have read the journal's policy and have the following competing interests: RM's doctoral work is funded by the Ligue Auvergne Rhône Alpes de Rugby à XIII. Part of this funding comes from a grant from the Association Nationale de la Recherche et de la Technologie (CIFRE agreement No. 2021/1144). This does not alter our adherence to PLOS ONE policies on sharing data and materials.

**Abbreviations:** 6MWT, 6-Meter Walking Test; AUC, Area Under the Curve; *c*-index, concordance index; BMI, Body Mass Index; CMI, Cognitive-Motor Interference; DT, Dual-Task; DTC, Dual-Task Cost; EF, Executive Functions; MoCA, Montreal Cognitive Assessment; ROC, Receiver Operating Characteristic; SRC-CES, Score de Risque de Chute des Centres d'Examen de Santé; ST, Single Task; TMT, Trail Making Test; W-TMT, Walking Trail Making Test.

of fall risk. Receiver operating characteristic analyses (providing the Area Under the Curve, AUC) were conducted on the same variables to evaluate and compare the capacity of both tasks to detect people at a high level of fall risk.

## Results

For the W-TMT, ANCOVA (corrected by spontaneous walking speed) showed significant group*condition interaction ($F_{4,194} = 8.27$, $p < .001$). *Post-hoc* analyses showed that, only in the W-TMT-B condition, execution time was longer for people at a high level of fall risk *vs* moderate ($M = 30.69s$, $SD = 15.45s$ *vs* $M = 21.54s$, $SD = 7.17s$, $p = .048$) and low levels ($M = 20.16s$, $SD = 7.90s$, $p = .031$). No differences were found for the 6MWT-DT execution times. The W-TMT-B better classified the levels of fall risk than the 6MWT-DT (*c*-index of respectively 0.63, 95%CI = .52−.74 *vs* 0.48, CI%95 = .38−.59, $p = .03$). The W-TMT-B also detected people at a high level of fall risk better than the 6MWT-DT (AUC of respectively 0.72, 95%CI = .53−.90 *vs* 0.46, 95%CI = .30−.62; $p = .03$) with a sensitivity of 61.54% and a specificity of 85.23%.

## Conclusions

Screening the level of fall risk with the W-TMT-B condition is more accurate than with the 6MWT-DT. The detection performance of the W-TMT-B condition could be explained by a high cognitive-motor interference mobilizing cognitive flexibility in a complex walking context. In contrast to *additional* DT, the *incorporated* feature of the W-TMT-B condition prevents any task prioritization strategy. Consequently, the W-TMT-B could be a simple and low-cost tool using only the execution time to detect people at a high level of fall risk in community-dwelling older people.

## Introduction

Falls are a major health issue for the older population worldwide. One-third of people aged over 65 and half of people aged over 80 experience at least one fall each year [1]. Falls often result in costly injuries and represent a heavy burden on societies that will increase in the future [2]. The multiple and interdependent fall risk factors increase the heterogeneity of levels of fall risk in older adults [3]. To reduce the occurrence of falls, individuals at a high level of fall risk need to be reliably identified because they are more likely to benefit from fall prevention interventions [4,5]. Numerous tools exist to identify people at a high level of fall risk, and can be divided into two main categories: self-reported questionnaires and performance-based measures [6]. Considering that no consensus has been reached on a unique tool, there is still a need to develop multifactorial assessments that can reliably screen older people who are at a high level of fall risk [7].

Older adults fall mostly while walking. Consequently, many fall risk assessment tools focus on locomotion performance, particularly in continuous, unperturbed straight-line walking [8,9]. A prospective study showed that a spontaneous walking

speed below the threshold of 0.7 m.s$^{-1}$ is associated with a high risk of falling [10]. However, the applicability of this threshold is limited in community-dwelling older people, because they typically have a higher spontaneous walking speed [11]. Using a complex walking task could make such assessments more relevant [12]. Complex walking involves changing direction, navigating on oriented pathways, or avoiding obstacles, which reflect everyday locomotion better than continuous, unperturbed straight-line walking [12,13]. Additionally, complexity is increased by requirement to divide attention between a walking task and a concurrent task, which particularly seems to increase fall occurrences in older people. People who struggle to maintain a conversation while walking are considered to be at a higher risk of falling than those who do not struggle [14]. Adding a cognitive task during simple walking (i.e., *additional* Dual-Task, DT) generally results in a slowing of walking speed [15]. This reflects the "Dual Task Cost" (DTC) (i.e., the cognitive cost of the concurrent cognitive task), and an increase in the DTC is associated with a high risk of falling [16,17].

Performance in *additional* DTs may depend on the prioritization strategy used by the participants (i.e., priority allocated to the locomotor or the *additional* cognitive task) because two distinct goals are present [18]. This prioritization strategy may result in inter-individual and intra-individual variations in performance that do not reflect the true probability of falling, thus limiting the relevance of using an *additional* DT to assess the level of fall risk. Moreover, studies suggest that the complexity of both the locomotor and cognitive tasks influences how relevant *additional* DTs are for fall risk assessment [12,19]. However, even when using an *additional* DT with a complex walking task, studies have obtained mixed results in predicting fall occurrences [20,21]. This may be due to the *additive* aspect of the DTs used, which still allows the participant to prioritize one task over the other [22]. Evidence also suggests that DT testing insufficiently detect people at a high level of fall risk with a sensitivity lower than 55.00% for different types of *additional* DT [23,24]. Thus, the most accurate type of DT remains to be determined, and alternative screening tools should be considered.

*Incorporated* cognitive-motor tasks are a promising alternative to *additional* DT, because the cognitive and motor tasks are interdependent. The single and shared overarching goal excludes any possibility of prioritization strategy [22]. Several *incorporated* cognitive-motor tasks involving complex walking have been developed based on neuropsychological assessments targeting fall risk-related factors [12]. Particularly, the Trail Walking Test, a walking task on a 5x5m square setup, has been designed based on the Trail Making Test (TMT) [25], a neuropsychological test assessing Executive Functions (EF) such as processing speed, visuospatial abilities, and cognitive flexibility [26]. As age-related decline in EF is a significant fall risk factor [27,28], this cognitive-motor task showed promising results in the screening of fall risk. A retrospective study found that the performance in the Trail Walking Test discriminated between fallers and non-fallers (based on history of fall) in community-dwelling older people [29]. A prospective study also found the Trail Walking Test more appropriate to predict falls than conventional mobility assessments (e.g., Timed Up and Go test and simple walking test) with a sensitivity of 66.10% and a specificity of 83.90% [30]. Thus, *incorporated* cognitive-motor tasks, designed based on the TMT test, can be considered particularly promising for predicting falls, as has been demonstrated in the case of the Trail Walking Test.

Another test derived from the TMT, the Walking Trail Making Test (W-TMT), has also been developed [31]. Compared to the Trail Walking Test, the W-TMT is quicker to set up and administer; it focuses on linear forward walking, requiring high precision in placing steps on targets while avoiding distractors, rather than on spatial navigation involving continuous changes in direction. The W-TMT consists of walking on targets following a cognitive rule in three separate conditions of increasing complexity: (i) W-TMT-N, that requires to walk on numeric targets in increasing order; (ii) W-TMT-A, the same as W-TMT-N but with distractors to avoid, and (iii) W-TMT-B, that requires to alternate between numeric and alphabetic targets (increasing and alphabetic order respectively) while avoiding distractors. Performance in the W-TMT is measured using execution time in each condition and Delta-WTMT (i.e., the cognitive cost of increased complexity between the B and A conditions). Interestingly, the execution time in the W-TMT-B condition and the Delta-WTMT were both significantly increased in people with EF-related cognitive impairments, a significant fall risk factor [31–33]. However, no study assessed whether the W-TMT performance could differentiate between different levels of fall risk in community-dwelling older people, compared to an *additional* DT involving simple walking.

The purpose of this retrospective study was to evaluate the detection performance of two different cognitive-motor tasks in the screening of a high level of fall risk in community-dwelling older people: the W-TMT, as an *incorporated* cognitive-motor task, and a simple walking DT, as an *additional* cognitive-motor task. Three hypotheses were formulated: (i) execution time in the W-TMT-B, the most complex condition, will be significantly higher for people at a high level of fall risk compared to people at lower levels; (ii) execution time in the W-TMT-B condition, will detect people at a higher level of fall risk better than chance; (iii) detection performance will be significantly better for the W-TMT-B than for the 6MWT-DT, which will perform no better than chance.

## Methods

All data used in this study were extracted from the 13EVAL cluster randomized controlled trial (Clinical Trials: NCT05625828). The objective of the 13EVAL study was to compare the effects of two different physical exercise programs on functional mobility and executive functions: (i) the "Silver XIII® Équilibre" program, a cognitive-motor program offered by the French Rugby-League Federation, and (ii) a control program based on walking and resistance training following the "Vivifrail®" international guidelines [34]. The pre-test and post-test assessment sessions included functional capacities, cognitive functions, and cognitive-motor abilities assessments. Data used in the present study were extracted only from the pre-test sessions.

### Participants

One hundred and one community-dwelling older people ($M = 75.88$ years, $SD = 6.23$, *range*: 65–92; 85 women) participated in the study. Participants were recruited by the French Rugby-League Federation between September 13th, 2022, and April 24th, 2025. Inclusion criteria were: being 65 years of age or older, living in the community, being able to walk independently, and to follow instructions for testing. Exclusion criteria were: more than three falls in the last 12 months, musculoskeletal disorders impairing posture or gait, central or peripheral neurological disease (e.g., previous stroke, Parkinson's disease), psychiatric disorders, probable depression (Geriatric Depression Scale score > 10) [35], neurocognitive disorder (Montreal Cognitive Assessment (MoCA) score < 18) [36], obesity (Body Mass Index (BMI) > 35), contraindication to practicing a physical activity, and involvement in another research protocol. If needed, participants were instructed to wear their common correction glasses.

### Ethics

The study was approved by a national ethic committee (CPP Île de France IV – EUDRACT 2022-001517-38). All participants were informed of the nature and aim of the study by an informative note and signed a written consent form. All procedures were in accordance with the Declaration of Helsinki with ethical standards, legal requirements and international norms.

### Measures

**General characteristics.** The duration of education (in years), age (in years), height (in m) and weight (in kg) were self-reported by the participants. BMI ((weight*height)/2, in kg.m$^{-2}$) was calculated.

**Global cognition.** The global cognition of the participants was measured using the MoCA (score range: 0–30) [36]. The MoCA includes measures of EF, language, attention, orientation, calculation, visuospatial ability, and memory. It enables the screening of early cognitive impairments in older adults (score greater than 18 and less than 26). MoCA scores were adjusted for educational level according to the recommendations of Nasreddine et al. (2005) [36].

**Cognitive flexibility.** Cognitive flexibility was assessed using the paper-pencil TMT [25]. This test consists of connecting the encircled numbers and letters disposed on an A4 paper sheet using a pencil. It includes two parts: part A, with only numbers from 1 to 25, and part B, in which participants must alternate between numbers from 1 to 12 and letters from A to L. Execution time was measured in each part using a hand chronometer, and a Delta-TMT was calculated as the difference between the execution times obtained in part B and part A. Execution times were compared to the normative standard of the TMT-A and TMT-B recorded in 8,995 French older adults living in the community by Amieva et al., (2009) [37]. These norms consider the person's gender, age, and level of education. Both the number of years of schooling and the attainment of the French Baccalaureate (Bac) were used to define the level of education. Using the provided percentiles, four normative classes were defined: deficient (below the 10th), inferior (between the 10th and 25th), average (from the 25th to the 75th), and superior (above the 75th).

**Level of fall risk.** The level of fall risk was assessed using a multifactorial screening tool recommended by the French National Health Insurance Fund and the French National Authority for Health: the "Score de Risque de Chute des Centres d'Examen de Santé" (SRC-CES) or Health Examination Center Fall Risk screening tool (translation proposed by Bongue et al. 2011) [38]. This tool was validated with a prospective design in a sample of 1,759 community-dwelling older people [39]. It includes a 5-item questionnaire and one performance measure, assessing a total of six fall risk factors. A score is assigned to each of them based on the following criteria (zero point for each criterion not met): being a woman (two points); living alone (one point); using psychoactive drug (one point); suffering from osteoarthritis (one point); number of fall(s) in the past year (two points for one fall, four points for two falls, six points for three and more falls); and having balance problems, intended as "a change in the arms position during the first five seconds of a one-leg balance test" (one point). The total score ranges from 0 to 12, and the authors provided cut-offs for three levels of fall risk: low (0–3), moderate (4–6), and high (7–12) [40]. In the present study, these three levels of fall risk were used for all analyses.

**The 6-meter walking test (6MWT).** Participants were instructed to walk straight at spontaneous speed in a 10-meter-long and 1-meter-wide corridor [41]. The initial 2-meter segment was for accelerating and reaching the spontaneous walking speed, while the last 2-meter segment was for decelerating and stopping (Fig 1). There were two experimental conditions: Single Task (ST), i.e., only walking at spontaneous speed, and DT, i.e., walking at spontaneous speed while performing a concurrent cognitive task. This cognitive task was a mental tracking task consisting of continuously subtracting seven starting from a random three-digit number provided by the investigator [18]. In the DT condition, instructions were given to perform the two tasks as best as possible. All participants performed two recorded trials in the ST condition and then one training trial, followed by two recorded trials in the DT condition. The execution time was measured for six meters using photocells Witty (Microgate©, Italy). For each condition, the mean of the two trials was used for analysis. The mean execution time in the ST condition was used to compute the spontaneous walking speed.

**The Walking – Trail Making Test (W-TMT).** Participants performed a version of the W-TMT inspired by Perrochon & Kemoun (2014) and adapted to fit with the 6MWT setup (Fig 1) [33]. The central measurement zone was 6-meter long and was prolonged by a 2-meter-long blank portion at the beginning and at the end. The targets were printed in the measurement zone in a "chevron-shape" disposition and measured 8 centimeters in diameter. The three experimental conditions of increasing complexity were: (i) W-TMT-N, with only numbers from 1 to 20 and where participants had to walk on the 20 numeric targets in increasing order; (ii) W-TMT-A, with the same 20 numbers as in W-TMT-N but with 20 other numeric distractors that the participants had to avoid while walking; (iii) W-TMT-B, with both numbers and letters as targets, where participants had to walk on the targets alternating between numbers and letters in increasing order for numbers and alphabetical order for letters (i.e., 1–A–2–B–3–C, etc.) while avoiding 20 other numeric and alphabetic distractors. To ensure comprehension of the instructions, participants were familiarized with each condition on a 3-meter-long and 1-meter-wide training carpet during at least one trial. During the test, an investigator stayed as close as possible

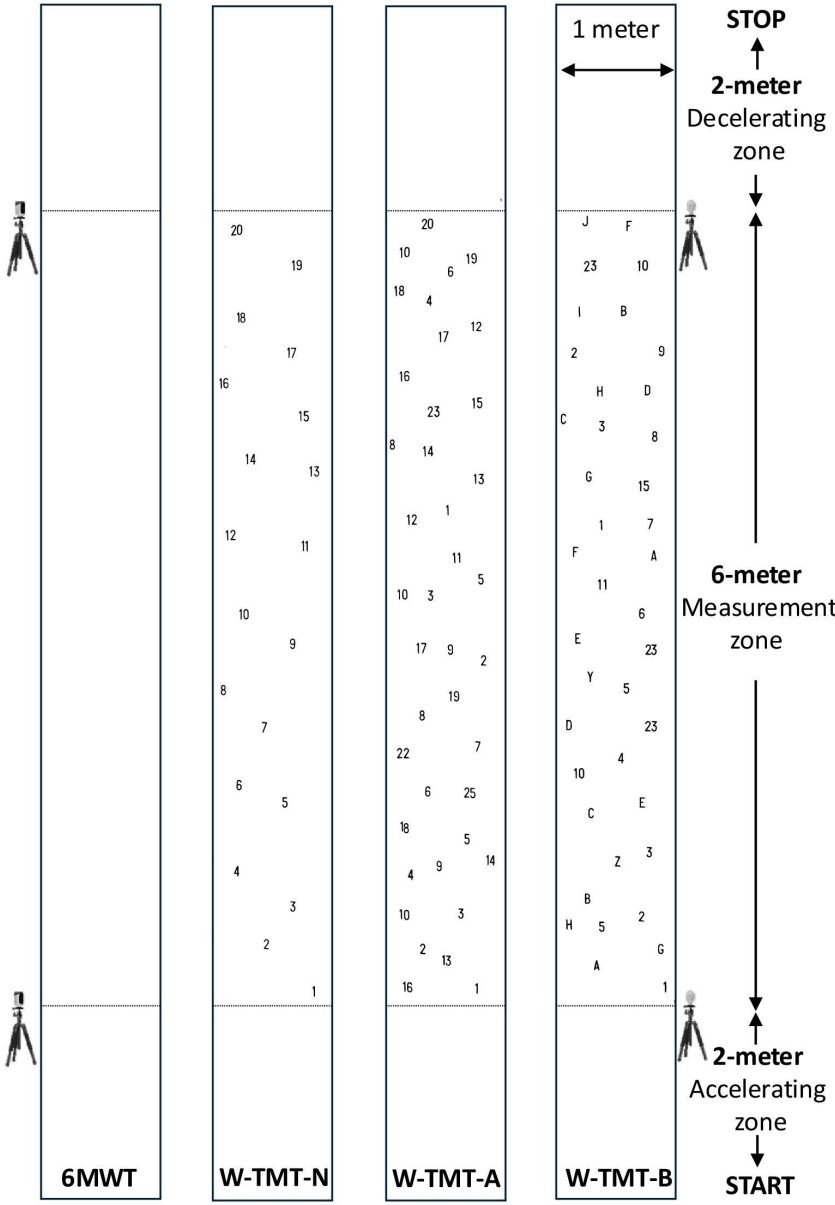

**Fig 1. The 6-Meter Walking Test (6MWT) and the Walking Trail Making Test (W-TMT) setups.**

to the participant to ensure safety. Participants started with the W-TMT-N condition, and the order of the W-TMT-A and W-TMT-B conditions was randomized. The execution time was measured for six meters using the same setup as in the 6MWT, and one trial was performed for each condition.

**Cognitive costs.** For the 6MWT, the cognitive cost was calculated using the following equation proposed by Kelly et al. (2010): DTC=[(DT- ST)/ST)]x100 [42]. For the W-TMT, the cognitive cost was calculated using the following equation proposed by Persad et al. (2008): Delta-WTMT=[(W-TMT-B–W-TMT-A)/W-TMT-A]x100 [32].

## Data analysis

Analyses were conducted using Jamovi® v2.6.17 (The Jamovi project, Sydney, Australia) and RStudio v2024.12.1 + 563 (R core team), with an alpha level of 0.05. Preliminary analyses were performed on dependent variables to check the normality of the distributions using the Kolmogorov-Smirnov test and the presence of outliers [43]. The Kolmogorov-Smirnov test was non-significant ($p > .05$) for all dependent variables, validating their normal distribution. Baseline between-group differences in age, weight, BMI, years of education, spontaneous walking speed, global cognition, and cognitive flexibility were compared using one independent factor "group" ANOVA. Chi-square tests were used to compare the distributions of different categorical variables across the three levels of fall risk: gender, number of previous falls, and normative performance classes on the TMT. For all ANOVA and ANCOVA, *post-hoc* comparisons were conducted using the Tukey correction when their results reached statistical significance. The effect sizes for all ANOVAs and ANCOVAs were reported using partial eta squared ($\eta^2_p$) for global and interaction effects and using Cohen's *d* for the *post-hoc* analysis. The following thresholds were selected for reporting effect sizes: (i) for $\eta^2_p$ .01−.06 small; .07−.14 medium; and >.14 large; and (ii) Cohen's *d* 0.2–0.5 small; 0.6–0.8 medium; and >0.8 large [44].

**Group performances in execution times and cognitive costs.** For the 6MWT, a 2-factor mixed ANOVA was conducted on execution time. The repeated factor "condition" consisted of the experimental condition (two levels: ST, DT). The independent factor "group" consisted of the level of fall risk (three levels: low, moderate, high). For the W-TMT, a 2-factor mixed ANCOVA was conducted on execution time, using spontaneous walking speed as a covariate. The repeated factor "condition" consisted of the experimental condition (three levels: W-TMT-N, W-TMT-A, W-TMT-B). The independent factor "group" consisted of the level of fall risk (three levels: low, moderate, high). For analysis of both the DTC and the Delta-WTMT, an independent factor "group" ANOVA was conducted for comparison among the three levels of fall risk.

**Concordance index (*c*-index) and Receiver Operating Characteristic analyses.** Execution time is a simple variable illustrating performance in the two cognitive-motor tasks. Considering that the most complex condition of each task will better detect people at a high level of fall risk, an evaluation of the detection performance of the 6MWT-DT and the W-TMT-B was performed based on the execution time and the level of fall risk.

*c-index analysis.* Due to the ordinal nature of the three levels of fall risk (i.e., low, moderate, and high), a *c*-index analysis was performed using the Hmisc R package [45]. This approach quantifies to what extent the W-TMT-B and the 6MWT-DT can classify individuals depending on their respective levels of fall risk. The *c*-index is the ratio between the number of concordant pairs and the total number of comparable pairs [46]. A comparable pair consists of two participants with different levels of fall risk. The pair is considered concordant if the participant with the highest level of fall risk also shows a higher execution time in the cognitive-motor task. Thus, the *c*-index reflects the probability that the execution time correctly classifies a randomly chosen higher risk level above a lower risk level. Interpretation of the *c*-index is as follows: a value of 1 indicates perfect classification (the model always classifies individuals with higher levels of fall risk above individuals with lower levels); a value of 0.5 corresponds to a model classifying randomly; and values between 0.5 and 1 reflect the extent to which the model performs better than chance. The *c*-index values were computed using a bootstrap resampling procedure with 5,000 iterations to provide 95% Confidence Intervals (95%CI) around the *c*-index estimates and to perform pairwise comparisons between the W-TMT-B and the 6MWT-DT classification performance.

*Receiver Operating Characteristic (ROC) analysis.* A focus was made on screening people at a high level of fall risk because of their higher need for fall prevention interventions than people at moderate and low levels. To apply a standard binary ROC analysis method, a dichotomization was made: the high level of fall risk category was compared to a group combining the low and moderate levels of fall risk category [47,48]. For each cognitive-motor task (6MWT-DT and W-TMT-B), the Area Under the Curve (AUC) was calculated. An AUC of 1 indicates perfect screening, whereas an AUC of 0.5 reflects chance-level screening. To statistically compare the detection performance of these two tasks, the AUCs were

compared using DeLong's test [49]. Global accuracy (in %), sensitivity (i.e., the true positive rate), and specificity (i.e., the true negative rate) were calculated. The optimal cutoff value for execution time was determined by Youden's index (sensitivity+(specificity-1)) [47].

## Results

### General characteristics

The average spontaneous walking speed was 1.14±0.19 m.s⁻¹; *range* [0.70–1.62]. The average MoCA score was 25.07±3.10; *range* [18–30], with 50.50% (N=51) of all participants with a MoCA score below the threshold of 26/30 suggesting a probable cognitive impairment. The average TMT part A execution time was 40.11±15.11 s; *range* [20.80–91.00], the average TMT part B was 92.46±48.10 s; *range* [35.70–300], and the average Delta-TMT was 52.34±40.59 s; *range* [5.00–243.13].

### Baseline differences

Regarding continuous variables, no differences were observed among the three levels of fall risk in demographic data (i.e., age, weight, and BMI), years of education, spontaneous walking speed, MoCA score, and both TMT-A and TMT-B execution times. Delta-TMT differed significantly among levels of fall risk ($F_{2,98}$ = 3.26, $p$ = .043, $\eta^2_p$ = .06, small effect). *Post-hoc* comparisons showed that people at a high level of fall risk had a larger Delta-TMT than people at a low level of fall risk (respectively $M$ = 73.64 s, $SD$ = 60.59, $vs$ $M$ = 41.23 s, $SD$ = 26.32 s, $p$ = .037, $d$ = 0.82, large effect) (Table 1).

Regarding categorical variables, gender ratios differed significantly, with proportionally more women than men in the high and moderate levels of fall risk than in the low level of fall risk ($\chi^2$ = 30.76, $p$ < .001). The distribution of non-fallers, single fallers, and recurrent fallers differed significantly among levels of fall risk ($\chi^2$ = 46.29, $p$ < .001). The high level of fall risk category is composed of only fallers, with proportionally more single fallers and recurrent fallers compared to the moderate and the low levels of fall risk (Table 2).

Table 1. Baseline differences in continuous variables among the three levels of fall risk given by the SRC-CES.

| Variable | Level of fall risk | | | $F_{2,98}$ | $p$ |
| --- | --- | --- | --- | --- | --- |
| | High (N=13) | Moderate (N=54) | Low (N=34) | | |
| Age (years) | 78.06 (±5.38) | 75.89 (±6.31) | 75.01 (±6.35) | 1.13 | .32 |
| Weight (kg) | 64.96 (±7.74) | 64.47 (±11.59) | 68.03 (±11.38) | 1.10 | .34 |
| BMI (kg.m⁻²) | 25.85 (±3.34) | 24.92 (±4.04) | 23.95 (±3.43) | 1.38 | .26 |
| Education (years) | 10.15 (±4.67) | 10.80 (±3.85) | 12.12 (±3.58) | 1.71 | .19 |
| Spontaneous Walking Speed (m.s⁻¹) | 1.09 (±0.12) | 1.12 (±0.19) | 1.18 (±0.21) | 1.63 | .20 |
| MoCA (score /30) | 25.23 (±3.42) | 25.02 (±3.24) | 25.09 (±2.84) | 0.02 | .98 |
| TMT-A (s) | 37.37 (±11.32) | 41.91 (±17.43) | 38.30 (±12.07) | 0.84 | .43 |
| TMT-B (s) | 111.01 (±68.80) | 96.13 (±49.38) | 79.53 (±32.32) | 2.42 | .09 |
| Delta-TMT (B-A, s) | 73.64 (±60.59)† | 54.21 (±40.35) | 41.23 (±26.32) | 3.26 | .043 |

Mean (±SD); MoCA scores adjusted for educational level; †Significant difference between high and low level in *post-hoc* comparisons.

**Table 2. Baseline differences in categorical variables among the three levels of fall risk given by the SRC-CES.**

| Variable | Level of fall risk | | |
| --- | --- | --- | --- |
| | High (N = 13) | Moderate (N = 54) | Low (N = 34) |
| Prevalence, % | 12.9 | 53.46 | 33.67 |
| Women, N (%) | 13 (100) | 53 (98.14) | 19 (55.88) |
| Non-fallers, N (%) | 0 (0) | 35 (64.81) | 32 (94.12) |
| Single fallers, N (%) | 8 (61.54) | 17 (31.48) | 2 (5.88) |
| Recurrent fallers, N (%) | 5 (38.46) | 2 (3.70) | 0 (0) |

*Note.* The number of fall(s) experienced by the participant in the past year was extracted from the SRC-CES results and quoted as: non-faller (0 fall), single faller (1 fall) and recurrent faller (2 or 3 falls).

### Execution time and cognitive cost in locomotor tasks

For the 6MWT, a significant main effect of the condition was found for the 2-factor mixed ANOVA ($F_{2,98} = 89.74$, $p < .001$, $\eta^2_p = .48$, large effect). *Post-hoc* analyses showed that participants took significantly more time to walk 6 meters in the DT condition than in the ST condition ($M = 10.74$ s, $SD = 4.79$, $vs$ $M = 5.44$ s, $SD = 0.98$, $p < .001$, $d = 1.17$, large effect). No interaction was found between conditions and the level of fall risk ($F_{2,98} = 0.64$, $p = .53$) (Fig 2A). The DTC analysis by levels of fall risk showed no difference ($F_{2,98} = 1.09$, $p = .34$) (Fig 2A).

For the W-TMT, a significant interaction between conditions and the level of fall risk was found ($F_{4,194} = 8.27$, $p < .001$, $\eta^2_p = .15$, large effect). *Post-hoc* comparisons showed that, in the W-TMT-B condition only, people at a high level of fall risk had a longer execution time than people at moderate ($M = 30.69$ s, $SD = 15.45$, $vs$ $M = 21.54$ s, $SD = 7.17$ s, $p = .048$, $d = 1.04$, large effect) and low levels of fall risk ($M = 20.16$ s, $SD = 7.90$, $p = .031$, $d = 1.19$, large effect) (Fig 2B). The Delta-WTMT analysis by level of fall risk reached statistical significance ($F_{2,98} = 8.74$, $p < .001$, $\eta^2_p = .15$, large effect). *Post-hoc* comparisons showed that people with a high level of fall risk had a greater percentage of slowing than people with moderate ($M = 138.22\%$, $SD = 85.61$, $vs$ $M = 69.06\%$, $SD = 45.42$, $p < .001$, $d = 1.24$, large effect) and low levels of fall risk ($M = 69.12\%$, $SD = 56.76$, $p < .001$, $d = 1.24$, large effect) (Fig 2B).

### c-index analyses

The execution time of the 6MWT-DT did not classify better than chance people depending on their respective level of fall risk ($c$-index = .48; IC%95 = .38−.59; $p = .73$). The execution time of the W-TMT-B correctly classified people according to their level of fall risk in 63.31% of the cases ($c$-index = .63; 95%CI = .52−.74; $p = .02$). Pairwise comparison between the 6MWT-DT and the W-TMT-B showed that the W-TMT-B had a significantly higher classification performance than the 6MWT-DT to classify people according to their level of fall risk based on execution time ($c$-index difference = .15; 95%CI = .01−.29; $p = .03$).

### ROC analyses

Screening of people at a high level of fall risk from those at lower risk levels was not better than chance based on the execution time of the 6MWT-DT (AUC = .46, 95%CI = .30−.62, $p = .61$). This same screening was correct based on the execution time of the W-TMT-B (AUC = .72, 95%CI = .53−.90, $p = .02$) (Fig 3 and Table 3). Pairwise comparison between the two tasks showed that the detection performance of people at a high level of fall risk was significantly higher for the W-TMT-B than for the 6MWT-DT (AUC difference = .26; 95%CI = .03−.49; $Z_{score} = 2.18$, $p = .03$). For the W-TMT-B condition, the overall accuracy was 82.18%, with a sensitivity of 61.54% and a specificity of 85.23% (Table 3). According to the Youden index, the best cutoff value was 28.17s.

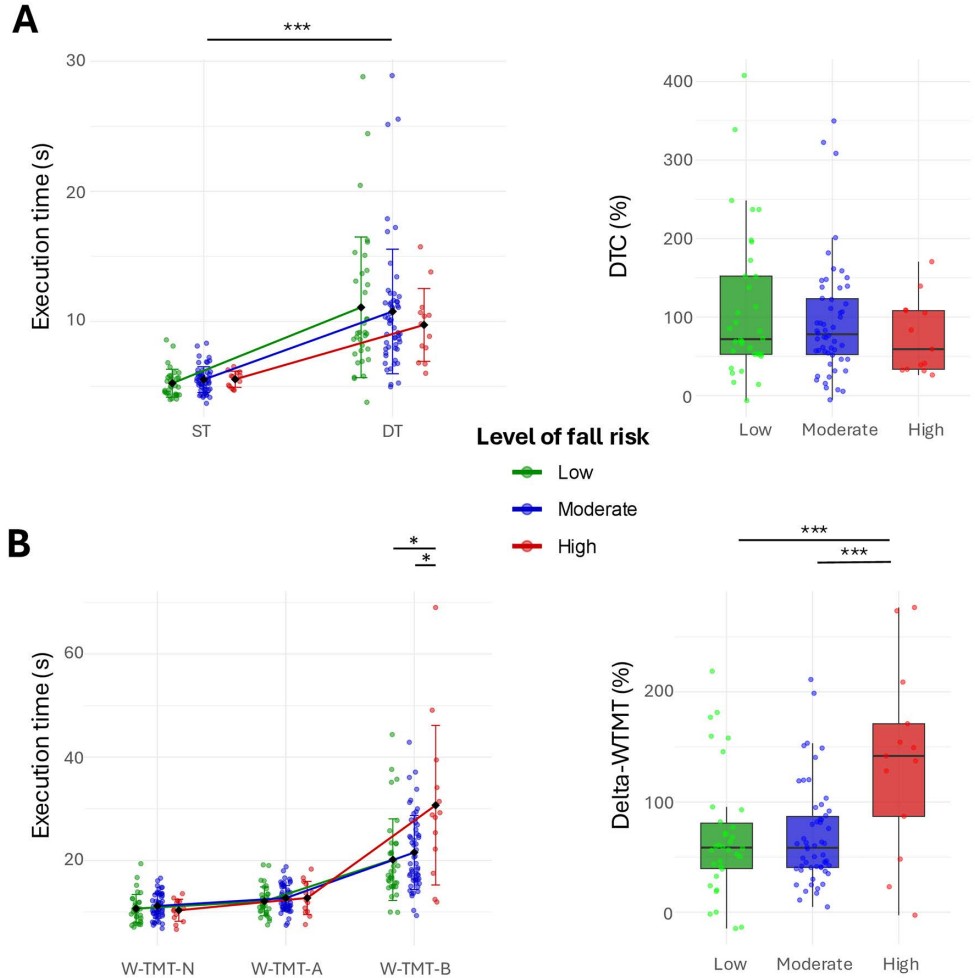

**Fig 2. Means and standard deviations of the 6MWT execution times and DTC (A), and of the W-TMT execution times and Delta-WTMT (B).**
(*$p < .05$; **$p < .01$; ***$p < .005$).

## Discussion

The purpose of this retrospective study was to evaluate the detection performance of two different cognitive-motor tasks in screening for a high level of fall risk in community-dwelling older people: the W-TMT, as an *incorporated* cognitive-motor task, and a simple walking DT, as an *additional* cognitive-motor task. Based on the *c*-index and AUC analyses, the main finding is that the execution time measured in the W-TMT-B condition is sensitive both to classify the levels of fall risk and to detect community-dwelling older people at a high level of fall risk from those at lower levels. Another important finding is that the detection performance was significantly higher for the W-TMT-B than for the 6MWT-DT, which performed no better than chance. This result suggests that *incorporated* cognitive-motor tasks are more accurate than *additional* cognitive-motor tasks for detecting people at a high level of fall risk among relatively healthy community-dwelling older adults.

### The W-TMT-B, a tool to detect older people at a high level of fall risk

The execution time measured in the W-TMT-B condition both classified the three levels of fall risk, based on the *c*-index, and detected people at a high level of fall risk from those at lower levels better than chance, based on the AUC (Fig 3).

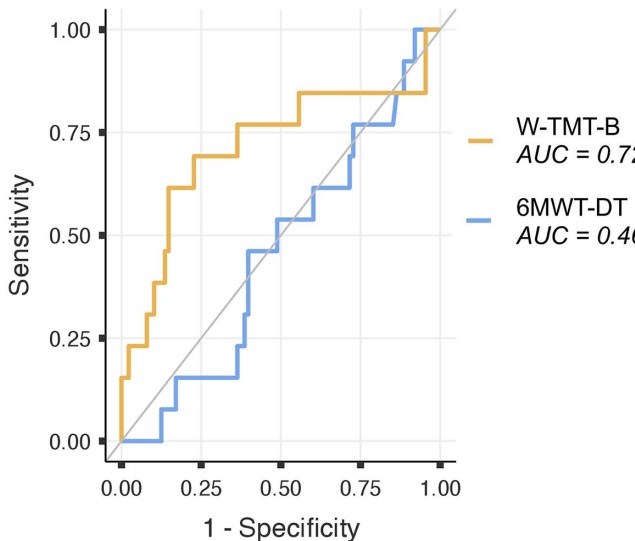

**Fig 3. Receiver operating characteristic curves for the 6MWT-DT and the W-TMT-B.**

**Table 3. Results of the receiver operating characteristic analysis.**

|  | 6MWT-DT |  | W-TMT-B |  |
| --- | --- | --- | --- | --- |
|  | Value | 95%CI | Value | 95%CI |
| **AUC (*p*)** | 0.46 (.61) | 0.30-0.62 | 0.72 (.02) | 0.53-0.90 |
| **Sensitivity (%)** | 100.00 | 75.29-100.00 | 61.54 | 31.58-86.14 |
| **Specificity (%)** | 7.95 | 3.26-15.70 | 85.23 | 76.06-91.89 |
| **Accuracy (%)** | 19.80 | 12.54-28.91 | 82.18 | 73.30-89.08 |

AUC: Area Under the Curve; 95%CI: 95 percent confidence interval

Specifically, a significant AUC of 0.72, a fair sensitivity (61.54%), and a high specificity (85.23%) was found (Table 3). The results of the present study are comparable to the results from the study of Bongue et al. (2011) (AUC of 0.70, a sensitivity of 19.20%, and a specificity of 96.5% for the cut-off of seven points determining the high level of fall risk category [39]), used to validate the SRC-CES fall risk screening tool with a prospective design. The present study also corroborates results of a prospective study based on the "Trail Walking Test" (AUC of 0.78, a sensitivity of 66.10% and a specificity of 83.90% in the B condition [30]). Thus, the W-TMT-B showed comparable accuracy, based on a measure of the level of fall risk. It is important to note that, in both previously cited studies and in the present study, the sensitivity (i.e., true positive rate) is lower than the specificity (i.e., true negative rate). This difference between sensitivity and specificity could be explained by the low prevalence of the category of interest (i.e., fallers or people at a high level of fall risk).

In the present study, the prevalence of the high level of fall risk category was only 12.90% (Table 2). Yet, all older people at a high level of fall risk were fallers. Among them 61.54% were single fallers and 38.46% were recurrent fallers (Table 2). This suggests that participants detected at a high level of fall risk by the W-TMT-B need to be oriented towards clinicians to provide a more precise assessment of fall risk factors and fall prevention interventions [5]. The high specificity indicates that the W-TMT-B performance is suitable for identifying people at low and moderate levels of fall risk. Consequently, people classified within these two categories could be oriented towards primary preventive care interventions. It

is crucial to point out that the best detection performance was found in the W-TMT most complex condition (W-TMT-B), which suggests this condition could be used to better detect older people at a high level of fall risk.

### The W-TMT-B condition involves cognitive flexibility, an independent fall risk factor

People at a high level of fall risk had the longest execution time during the W-TMT-B condition (Fig 2B). This could be explained by early EF alteration in this population. Indeed, Persad et al. (2008) showed that older people with early EF impairment had a significantly longer execution time in the W-TMT-B condition compared to healthy older people, and this result was replicated with all versions of the task [32,33,50,51]. Moreover, falls are common in cognitively impaired older people and particularly in those with EF impairment [28,52–54]. EF are also known to be highly involved in quick adaptation to changes, especially when a fast and adequate reaction to postural disturbances is required to avoid falling [55–57]. Thus, elongation of the execution time in the W-TMT-B condition might reflect early alteration of EF in community-dwelling older people and help to classify these people at a high level of fall risk. The Delta-WTMT focuses on the cognitive functions specifically mobilized during the W-TMT-B condition, when compared to the W-TMT-A condition. The particularity of the W-TMT-B condition is the alternating sequence of numbers and letters, as in the original paper-pencil TMT part B [25]. Since the TMT part B condition is associated with cognitive flexibility, so should the W-TMT-B condition [26]. Consequently, the Delta-WTMT scores might reflect the participants' efficiency in cognitive flexibility. As the participants at a high level of fall risk also showed a significantly higher paper-pencil Delta-TMT than participants at a low level (Table 1), participants at a high level should have the greatest cognitive flexibility impairments. Mobilization of cognitive flexibility during the W-TMT-B condition can be further explained by the goal-oriented locomotor processes that take place during the execution of the task. The complex walking path induced by the placement of targets in a non-linear "chevron-shaped" pattern reproduces challenging terrain conditions such as an obstacle avoidance situation [31]. These tasks mobilize cognitive flexibility to adapt the goal-oriented locomotor processes continuously [58]. Consequently, performing the W-TMT seems to mobilize cognitive flexibility because it implies a complex conscious control, including evaluation of the spatial environment and efficient motor planning [8].

### Screening with the W-TMT-B is more accurate than with the 6MWT-DT

A novelty of the present study is that both diagnostic accuracy analyses showed that the W-TMT-B was significantly more performant than the 6MWT-DT to (i) classify people according to the three levels of fall risk, based on the *c*-index, and (ii) detect people at a high level of fall risk from the two other levels, based on the AUC (Fig 3 and Table 3). These results offers new insights to cope with the conclusions of different reviews suggesting that *additional* DT tasks showed mixed results in the prediction of falls [23,24,59,60]. Cognitive-Motor Interference (CMI) taking place in a DT situation requires that both task must be performed simultaneously [18].

In an *additional* DT, the prioritization strategy offers a solution to reduce the level of CMI by giving priority to one task over the other. In the DT condition of the 6MWT, participants were instructed to perform both tasks as well as possible, leaving the possibility to prioritize one objective over the other [22]. Consequently, the performance in this condition depends on factors that could orient task prioritization. Participants may have prioritized based on their own estimation of the complexity of the cognitive task. In this study, a mental-tracking arithmetic task was used, and results showed a larger increase of execution time in the DT condition compared to previous studies with similar tasks and populations [18]. Considering the population tested in the present study, this finding confirms that the mental-tracking task was of sufficiently high cognitive complexity for the DT. The motor task consisted of straight walking at spontaneous speed, which primarily relies on functional capacities [61,62]. Given that most participants had a high spontaneous walking speed, the complexity of the locomotor task was probably too low [10,62,63]. Consequently, the combination of a low complexity in the motor task with a high complexity in the cognitive task may have led participants to prioritize the cognitive task. The review

conducted by Yogev-Seligmann et al. (2012) about *additional* DTs has reported that community-dwelling older people, as well as young adults, often adopt a "posture-second" strategy to focus more on the cognitive task due to a high confidence in their postural control [64].

Another possibility is that participants may have prioritized depending on their level of cognitive reserve [65]. Half of the participants had a MoCA score below 26, which is the threshold to detect early cognitive impairments [36]. Considering that people with poor cognitive performance may be the ones with a higher level of fall risk, they could have been detected by a longer execution time. However, people with probable cognitive impairment may have prioritized the locomotor task due to their reduced ability to overcome the high cognitive complexity of the DT condition, resulting in a short execution time. This could explain why no differences were found in 6MWT-DT between people at high level of fall risk compared to lower levels (Fig 2A).

In contrast, no task prioritization is possible during the W-TMT-B because the cognitive and motor tasks are interdependent and share a single overarching goal. Unlike the *additional* DT paradigm, CMI takes place in an *incorporated* DT paradigm as long as the participant is performing the task [66]. Moreover, by incorporating a complex cognitive rule into a complex walking paradigm, it possibly mobilizes shared neural networks for processing both the cognitive and motor dimensions of the task increasing the level of CMI [66–68].

## Limitations and perspectives

Since this is an ancillary study, the results were obtained using a retrospective design based on the pre-test outcomes of the 13EVAL study. As a result, the predictive value of the assessment needs to be determined. Moreover, the 13EVAL study excluded individuals who had experienced more than three falls to ensure their safety during physical activity interventions. Recurrent fallers with more than three falls could have been classified in the high level of fall risk category. As a result, it could have restricted the number of participants in this category. In addition, the sample consisted of 81.17% (N = 85) women, which limits the generalization of results to the men population. More research using a prospective design with a follow-up and reliability analyses is needed to enhance the clinical value of this assessment and determine whether the W-TMT-B can become an effective fall prediction tool.

## Conclusions

The W-TMT-B execution time efficiently detected people at a high level of fall risk compared to lower levels in community-dwelling older people. The detection performance of the W-TMT-B condition could be explained by a high cognitive-motor interference induced by the involvement of cognitive flexibility in a complex walking context. In contrast, the poor accuracy of the DT assessment during the 6MWT could be explained by the low complexity of the walking task proposed in an *additional* context that left open access to task prioritization strategies. Together, the findings of this study support that *incorporated* cognitive-motor tasks proposed in complex walking contexts, such as the W-TMT, could be more appropriate than *additional* DTs based on simple walking to investigate the level of fall risk in community-dwelling older people. Consequently, with a simple variable (i.e., the execution time) and a cut-off score of 28.17s, the W-TMT-B condition could be a low-cost and operational tool easily administered by physical activity instructors to: (i) propose primary preventive care interventions to older adults at low and moderate levels of fall risk; and (ii) refer people at high level of fall risk for in-depth clinical investigations.

## Supporting information

**S1 Results. Normative classes for the Trail Making Test.**
(DOCX)

## Acknowledgments

The authors would like to express their sincere gratitude to the Ligue Auvergne Rhône Alpes de Rugby à XIII for its financial and material support, as well as for its invaluable assistance in recruiting participants for this study. We would also like to thank its president, Mr. Jacques CAVEZZAN, for his enthusiasm and unconditional support throughout Rafael MAUTI's doctoral project.

## Author contributions

**Conceptualization:** Pascal Chabaud.

**Data curation:** Rafael Mauti, Pascal Chabaud.

**Formal analysis:** Rafael Mauti.

**Funding acquisition:** Pascal Chabaud.

**Investigation:** Rafael Mauti, Mélanie Gallot, Arnaud Peronin.

**Methodology:** Rafael Mauti.

**Project administration:** Pascal Chabaud.

**Supervision:** Patrick Fargier, Pascal Chabaud.

**Visualization:** Rafael Mauti.

**Writing – original draft:** Rafael Mauti, Romain Tisserand, Patrick Fargier, Pascal Chabaud.

**Writing – review & editing:** Rafael Mauti, Romain Tisserand, Anaïck Perrochon, Mélanie Gallot, Thomas Gilbert, Patrick Fargier, Pascal Chabaud.

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
