## [Decision Letter · Decision Letter 0]

2 Apr 2026

PONE-D-25-65198The Walking Trail Making Test is more accurate than a Dual-Task Walking Test for screening the level of fall risk among community-dwelling older peoplePLOS One

Dear Dr. Chabaud,

Thank you for submitting your manuscript to PLOS ONE. After careful consideration, we feel that it has merit but does not fully meet PLOS ONE’s publication criteria as it currently stands. Therefore, we invite you to submit a revised version of the manuscript that addresses the points raised during the review process.

Respond to the reviewer's comments in your revised manuscript.==============================

We look forward to receiving your revised manuscript.

Kind regards,

Michael Francis Salvatore

Academic Editor

PLOS One

Journal Requirements:

“This research was funded by the Ligue Auvergne Rhône Alpes de Rugby à XIII and the National Association of Research and Technology by the grant obtained under CIFRE agreement no. 2021/1144.”

“This research was funded by the Ligue Auvergne Rhône Alpes de Rugby à XIII and the National Association of Research and Technology by the grant obtained under CIFRE agreement no. 2021/1144.”

We note that one or more of the authors is affiliated with the funding organization, indicating the funder may have had some role in the design, data collection, analysis or preparation of your manuscript for publication; in other words, the funder played an indirect role through the participation of the co-authors. If the funding organization did not play a role in the study design, data collection and analysis, decision to publish, or preparation of the manuscript and only provided financial support in the form of authors' salaries and/or research materials, please do the following:

1. Review your statements relating to the author contributions, and ensure you have specifically and accurately indicated the role(s) that these authors had in your study. These amendments should be made in the online form.

2. Confirm in your cover letter that you agree with the following statement, and we will change the online submission form on your behalf:

“The funder provided support in the form of salaries for authors [insert relevant initials], but did not have any additional role in the study design, data collection and analysis, decision to publish, or preparation of the manuscript. The specific roles of these authors are articulated in the ‘author contributions’ section.

Reviewers' comments:

Reviewer's Responses to Questions

**Comments to the Author**

1. Is the manuscript technically sound, and do the data support the conclusions?

Reviewer #1: Yes

2. Has the statistical analysis been performed appropriately and rigorously? 

Reviewer #1: Yes

3. Have the authors made all data underlying the findings in their manuscript fully available?

Reviewer #1: Yes

4. Is the manuscript presented in an intelligible fashion and written in standard English?

Reviewer #1: Yes

5. Review Comments to the Author

Reviewer #1: The manuscript reflects a very interesting and important topic to investigate to promote primary care physicians to include testing for fall risks in their senior patients. This study was thoughtfully designed with scientific rigor using appropriate as well as accurate data analyses and interpretations. Only very minor edits to make perhaps during the editorial process.

6. PLOS authors have the option to publish the peer review history of their article (what does this mean?). If published, this will include your full peer review and any attached files.

Reviewer #1: **Yes:** Vicki A. Nejtek-Salvatore

---

## [Author Response · Author response to Decision Letter 1]

21 Apr 2026

Given the large number of tables in our responses to the editor’s and reviewer’s comments, please refer to the Word file named "02 - Response_to_Reviewers".

---

## [Decision Letter · Decision Letter 1]

23 Apr 2026

The Walking Trail Making Test is more accurate than a Dual-Task Walking Test for screening the level of fall risk among community-dwelling older people

PONE-D-25-65198R1

Dear Dr. Chabaud,

We’re pleased to inform you that your manuscript has been judged scientifically suitable for publication and will be formally accepted for publication once it meets all outstanding technical requirements.

Kind regards,

Michael Francis Salvatore

Academic Editor

PLOS One

Additional Editor Comments (optional):

Dear Dr. Chabaud,

Congratulations on your efforts to produce a solid study and manuscript. As stated by the reviewer, this study provides critical information to the field for those interesting in assessment of motor impairments, and adds to our knowledge about the relationship of cognitive function to motor capabilities.

All the best,

Michael

Reviewers' comments:

Reviewer's Responses to Questions

**Comments to the Author**

1. If the authors have adequately addressed your comments raised in a previous round of review and you feel that this manuscript is now acceptable for publication, you may indicate that here to bypass the “Comments to the Author” section, enter your conflict of interest statement in the “Confidential to Editor” section, and submit your "Accept" recommendation.

Reviewer #1: All comments have been addressed

2. Is the manuscript technically sound, and do the data support the conclusions?

Reviewer #1: Yes

3. Has the statistical analysis been performed appropriately and rigorously? 

Reviewer #1: Yes

4. Have the authors made all data underlying the findings in their manuscript fully available?

Reviewer #1: Yes

5. Is the manuscript presented in an intelligible fashion and written in standard English?

Reviewer #1: Yes

6. Review Comments to the Author

Reviewer #1: The authors have made all of the revisions and have added increased awareness concerning the importance of their study. This effort on the author's part should highly influence primary care physicians and other clinicians to include an integrated cognitive-motor test such as that described in this excellent manuscript. Providing the additional tables and information concerning cut-offs and reference ranges for fall risks are especially helpful. It is not often that I am given such a high quality of work to review. I hope that these researchers continue to explore additional modalities portending to risks of decline in our elder generation that will benefit patients, their caregivers, and their treating physicians.

7. PLOS authors have the option to publish the peer review history of their article (what does this mean?). If published, this will include your full peer review and any attached files.

Reviewer #1: **Yes:** Vicki A. Nejtek, Ph.D.

---

## [Editor Report · Acceptance letter]

PONE-D-25-65198R1

PLOS One

Dear Dr. Chabaud,

I'm pleased to inform you that your manuscript has been deemed suitable for publication in PLOS One. Congratulations! Your manuscript is now being handed over to our production team.

Kind regards,

on behalf of

Dr. Michael Francis Salvatore

Academic Editor

PLOS One